# Pharmacokinetic Feasibility of Stability-Enhanced Solid-State (SESS) Tenofovir Disoproxil Free Base Crystal

**DOI:** 10.3390/pharmaceutics15051392

**Published:** 2023-05-01

**Authors:** Byoung Hoon You, Mingoo Bae, Seung Yon Han, Jieun Jung, Kiwon Jung, Young Hee Choi

**Affiliations:** 1College of Pharmacy and Integrated Research Institute for Drug Development, Dongguk University_Seoul, 32 Dongguk-lo, Ilsandong-gu, Goyang-si 10326, Gyeonggi-do, Republic of Korea; hoon4131@nate.com (B.H.Y.); nophra88@naver.com (M.B.); hsyglory@gmail.com (S.Y.H.); wldms5007@naver.com (J.J.); 2College of Pharmacy, CHA University, 120 Haeryong-ro, Pocheon-si 13488, Gyeonggi-do, Republic of Korea; 3Oncobix Co., Ltd., 120, Heungdeokjungang-ro, Giheung-gu, Yongin-si 16950, Gyeonggi-do, Republic of Korea

**Keywords:** tenofovir (TEV), stability-enhanced solid-state TD free base crystal (SESS-TD crystal), tenofovir disoproxil fumarate (TDF), absorption, bioavailability

## Abstract

Tenofovir (TEV) is a nucleotide reverse transcriptase inhibitor used against human immunodeficiency virus (HIV) reverse transcriptase. To improve the poor bioavailability of TEV, TEV disoproxil (TD), an ester prodrug of TEV, was developed, and TD fumarate (TDF; Viread^®^) has been marketed due to the hydrolysis of TD in moisture. Recently, a stability-enhanced solid-state TD free base crystal (SESS-TD crystal) was developed with improved solubility (192% of TEV) under gastrointestinal pH condition and stability under accelerated conditions (40 °C, RH 75%) for 30 days. However, its pharmacokinetic property has not been evaluated yet. Therefore, this study aimed to evaluate the pharmacokinetic feasibility of SESS-TD crystal and to determine whether the pharmacokinetic profile of TEV remained unchanged when administering SESS-TD crystal stored for 12 months. In our results, the *F* and systemic exposure (i.e., AUC and *C*_max_) of TEV in the SESS-TD crystal and TDF groups were increased compared to those in the TEV group. The pharmacokinetic profiles of TEV between the SESS-TD and TDF groups were comparable. Moreover, the pharmacokinetic profiles of TEV remained unchanged even after the administration of the SESS-TD crystal and TDF stored for 12 months. Based on the improved *F* after the SESS-TD crystal administration and the stable condition of the SESS-TD crystal after 12 months, SESS-TD crystal may have enough pharmacokinetic feasibility to replace TDF.

## 1. Introduction

Tenofovir (TEV), a nucleotide analog of adenosine 5′-monophosphate, is a reverse transcriptase inhibitor against human immunodeficiency virus (HIV) reverse transcriptase [1]. Due to its poor bioavailability, Gilead Sciences Inc. developed tenofovir disoproxil (TD) as an ester prodrug of TEV [2], which is marketed as tenofovir disoproxil fumarate (TDF; Viread^®^) [3,4]. TD hydrolyzes and forms formaldehyde in contact with moisture, leading to the formation of TD dimer [5,6,7]. After oral administration of TDF, it is rapidly hydrolyzed to TEV, which is then converted intracellularly into TEV diphosphate, a pharmacologically active metabolite [3,4,8]. In other words, TDF improves the absorption of TEV in the gastrointestinal tract and its influx into cells compared to the administration of TEV itself [8].

However, there are drawbacks in the physiochemistry and manufacturing stages of TDF. TDF is a hydrophilic salt and is easily degraded in the presence of moisture. TD was developed in oil form, which is inconvenient for the manufacturing process of TDF [9,10]. Although the syntheses of TD and TDF crystals have been attempted, their crystal structures have not been fully characterized [4,11,12]. An et al. (2017) synthesized a new TD free base crystal with improved hygroscopic properties (i.e., stability-enhanced) in the solid state; the solubilities of the stability-enhanced solid-state TD free base crystal (SESS-TD crystal) in aqueous solution and under gastrointestinal pH conditions were comparable to TDF, and the stability of the SESS-TD crystal under accelerated conditions (40 °C, RH 75%) was better than that of TDF [11]. The crystalline structure can influence its physicochemical properties, such as the solubility, stability, and dissolution rate, and its pharmacokinetic profiles. Therefore, the pharmacokinetic feasibility of SESS-TD crystal should be evaluated by comparing the TEV pharmacokinetics between the SESS-TD crystal and TDF administrations.

## 2. Materials and Methods

### 2.1. Chemicals

The TEV (PMPA; purity 98.00%), synthesized stability-enhanced solid state TD free base crystal (SESS-TD crystal; purity 99.93%), and TDF (i.e., TDF form-I; purity 98.37%) (Figure 1) were provided by Dr. Kiwon Jung (CHA University and Oncobix Co., Ltd., Gyeonggi-do, Republic of Korea) [11]. The physicochemical properties of the SESS-TD crystal are described in Table 1. Paclitaxel (internal standard (IS) of liquid chromatography–tandem mass spectrometry (LC-MS/MS)) was purchased from Sigma-Aldrich (St. Louis, MO, USA). Acetonitrile and water were purchased from Honeywell (St Charlotte, NC, USA), and formic acid was purchased from Wako Chemicals (Chuo-Ku, Osaka, Japan). All other chemicals and reagents used were of analytical grade.

### 2.2. Animals

The animal study protocols were approved by the Institute of Laboratory Animal Resources of Dongguk University_Seoul, Korea (approval no. IACUC-2019-025 and IACUC-2021-041). Male Sprague–Dawley rats (5–7 week-old, weighing 150–250 g) were purchased from the Charles River Company Korea (Orient, Gyeonggi-do, Republic of Korea) and acclimated for one week prior to the start of the study. All rats were given free access to food and water and were randomly housed with two per cage under strictly controlled environmental conditions (20–25 °C and 48–52% relative humidity). A 12 h light/12 h dark cycle with an intensity of 150 to 300 Lux was maintained.

### 2.3. LC-MS/MS Analysis of TEV in TEV, SESS-TD Crystal, and TDF Samples

An analysis of TEV in biological samples was performed by modifying the previously reported methods [17,18,19]. The API 4000 triple quadrupole mass spectrometer system (Framingham, MA, USA) was used for all analyses. The multi-reaction monitoring (MRM) mode with an electrospray ionization (ESI) interface was used for positive ions ([M+H])^+^ at a capillary voltage of 5500 V, a desolvation gas temperature of 500 °C, a nebulizing gas flow of 50 L/min, a turbo ion-spray gas flow of 50 L/min, a curtain gas flow of 20 L/min, a ring voltage of 5500 V, and collision gas (nitrogen) pressure of 5 Torr. The m/z values were 288.330 → 176.100 and 854.222 → 286.100 for TEV and IS, respectively. Chromatographic separation was performed using a reversed-phase C_18_ column (Waters X-select C_18_, 2.1 mm × 100 mm i.d., 3.5 µm particle size; Waters) at a flow rate of 0.4 mL/min. The mobile phase was composed of 0.1% formic acid in water (A) and 0.1% formic acid in acetonitrile (B). The gradient elution was performed using the mobile phase with an A/B ratio of 90:10 (*v*/*v*) initially, which was changed to a ratio of 10:90 (*v*/*v*) at 3.5 min, and then returned to the initial composition at 6.1 min, which was then maintained for 7 min (i.e., the total run time). The analytical data were processed using Analyst software 1.7.3 (AB Sciex, Framingham, CA, USA).

To prepare the working solutions and biological standard samples of TEV, stock solutions of TEV in DMSO were prepared at a concentration of 20 mg/mL and then serially diluted by methanol. The working solution of TEV was added to drug-free plasma samples to achieve final concentrations of 0.01, 0.025, 0.1, 0.5, 1, 2, 5, 10, 20, and 200 µg/mL. To deproteinize a 50 µL biological sample, 200 µL of acetonitrile containing 1 µg/mL of IS was added. The sample was vortexed and centrifuged for 10 min at 13,000 rpm, and 10 µL of the supernatant was injected into the column. The peaks of TEV and IS were obtained, and their retention times were 0.92 and 4.40 min, respectively (Appendix A). The limit of detection for TEV in the biological standard samples of plasma, urine, and GI were 0.002, 0.2, and 0.02 µg/mL, respectively. The calibration curve ranges for TEV in the biological standard samples of plasma, urine, and GI were 0.01–200 µg/mL (y = 472 x + 1645, R^2^ = 0.999), 1–20 µg/mL (y = 17.3 x + 1143, R^2^ = 1.00), and 0.1–20 µg/mL (y = 59.5 x + 1065, R = 0.998), respectively. The calibration curves for TEV in the biological standard samples were obtained from their peak area ratios relative to those of the IS by linear regression.

### 2.4. Rat Plasma Protein Binding of TEV in Rat Plasma Spiked with TEV, the SESS-TD Crystal, or TDF

The rat plasma protein binding values of TEV from TEV, the SESS-TD crystal, and TDF were determined using a rapid equilibrium dialysis (RED) device (Thermo Fisher Scientific, Waltham, MA, USA) with a molecular weight cutoff of 8.0 KDa, following previously reported methods [20,21]. Briefly, 100 µL of fresh rat plasma containing 0.3 µg/mL TEV, 0.543 µg/mL SESS-TD crystal, and 0.664 µg/mL TDF (at final concentrations equivalent to 0.3 µg/mL TEV) were added to the plasma chamber, and 300 µL of dialysis buffer solution (i.e., 0.1 M phosphate buffered saline) was added to the buffer chamber. The samples were incubated for 4 h at 37 °C with stirring at 250 rpm, after which a 50 µL sample from each chamber was collected and stored at –80 °C for subsequent LC-MS/MS analysis of TEV.

### 2.5. TEV Pharmacokinetics after Administration of TEV, SESS-TD Crystal, or TDF in Rats

Prior to the experiment, the rats were fasted overnight but allowed access to water. On the day of the experiment, the rats were exposed to anesthetic doses of ethyl ether by inhalation for 5 min. Cannulation of the jugular vein (for intravenous drug administration only in the intravenous study) and the carotid artery (for blood sampling in both the intravenous and oral studies) was performed using methods similar to those reported previously [20,21]. In the blood sampling, the total blood sampling volume was 0.96–1.08 mL (i.e., 0.12 mL of blood was collected 9 and 8 times in the intravenous and oral studies, respectively), which is acceptable considering that 1.7–2.1 mL/300 g without fluid supplement and 2.5–3.2 mL/300 g with fluid supplement are acceptable in rats [22].

For the intravenous study, a dose of 20 mg (4 mL)/kg of TEV (dissolved in distilled water) was intravenously administered to the rats. Approximately 0.12 mL of blood was collected via the carotid artery at 1, 5, 15, 30, 60, 120, 240, 360, and 480 min after TEV administration. The blood sample was immediately centrifuged at 13,000 rpm and 4 °C for 10 min, and 50 μL of the supernatant (i.e., plasma) was collected. At the end of 24 h, the urine sample was collected by rinsing each metabolic cage with 20 mL of distilled water, which was mixed with the urine collected over the previous 24 h. At this time, the rats were then sacrificed by cervical dislocation, and the gastrointestinal tract, including its contents and feces, was removed, transferred into a beaker, and cut into small pieces. One hundred milliliters of methanol was added to the beaker to extract the TEV, and the mixture was manually shaken. A 50 μL aliquot of the supernatant was collected from each beaker and stored at –80 °C (Western Mednics, Revco ULT 1490 D–N-S) prior to the LC-MS/MS analysis of TEV.

For the oral study, the rats were orally administered TEV, the SESS-TD crystal, and TDF (dissolved in distilled water) at a dose of 20 mg (4 mL)/kg of TEV using a gastric gavage tube after overnight fasting with free access to water. Blood samples of approximately 0.12 mL were collected via the carotid artery at 5, 15, 30, 60, 120, 240, 360, and 480 min after the oral administration of TEV, the SESS-TD crystal, and TDF. The other procedures in the oral study were similar to those in the intravenous study. All biological samples were stored at −80 °C (Western Mednics, Revco ULT 1490 D–N-S, Asheville, NC, USA) prior to the LC-MS/MS analysis of TEV.

### 2.6. TEV Pharmacokinetics after Administration of the SESS-TD Crystal and TDF Stored for 12 Months in Rats

For the SESS-TD crystal and TDF, oral administration of the SESS-TD crystal or TDF was performed after storage at room temperature for 12 months. The intravenous and oral administrations of TEV were conducted as described above to serve as the positive control. The other procedures for the pharmacokinetic studies were the same as those described above.

### 2.7. Pharmacokinetic Parameters

The pharmacokinetic parameters, including the total area under the plasma concentration–time curve from time zero to infinity (AUC), the terminal half-life (*t*_1/2_), the apparent volume of distribution at a steady state (*V*_ss_), the total body clearance (CL), the renal clearance (CL_R_), the non-renal clearance (CL_NR_), and the apparent total body clearance after oral administration (CL/*F*), were calculated using non-compartmental analysis (WinNonlin; Pharsight Corporation and PK solver, version 2.1; Scientific, Sunnyvale, CA, USA). The absolute *F* of the TEV, SESS-TD crystal, and TDF was estimated by dividing the oral AUC of each compound by the intravenous AUC of TEV. The relative *F* of the SESS-TD crystal and TDF was estimated by dividing the oral AUC of each compound by the oral AUC of TEV. The CL_oral_ was estimated by multiplying CL/*F* with the absolute *F*. The early peak plasma concentration (*C*_max_) and time to reach *C*_max_ (*T*_max_) were obtained directly from the plasma concentration–time data. The mean ‘true’ unabsorbed fractions (‘*F*_unabs_’) were estimated using a reported equation [23]: oral GI_24h_ = ‘*F*_unabs_’ + (intravenous GI_24h_ × absolute *F*).

### 2.8. Statistical Analysis

A *p* value < 0.05 was deemed to be statistically significant using Tukey’s multiple range test with the Statistical Package of Social Sciences (SPSS) *posteriori* analysis of variance (ANOVA) among the three means for the unpaired data. All data are expressed as the mean ± standard deviations except *T*_max_, which is expressed as the median (ranges).

## 3. Results

### 3.1. Rat Plasma Protein Binding of TEV

The rat plasma protein binding values of TEV were 1.06 ± 0.579%, 0.958 ± 0.0845%, and 1.14 ± 0.230% when TEV, the SESS-TD crystal, and TDF (equivalent to 0.3 µg/mL TEV), respectively, were spiked to the rat plasma. There were no significant differences among the three groups.

### 3.2. TEV Pharmacokinetics after Administration of TEV, the SESS-TD Crystal, and TDF in Rats

The mean arterial plasma concentration–time profiles and relevant pharmacokinetic parameters of TEV after the intravenous and oral administration of TEV are shown in Figure 2 and Table 2, respectively. Following intravenous administration, the TEV was rapidly and extensively distributed throughout the body, with a volume of distribution of 1917 mL/kg. The *t*_1/2_ of TEV was 166 min, and it was predominantly eliminated by renal excretion, with 65.9% of the administered TEV dose excreted unchanged in the urine. Biliary excretion of the TEV was likely negligible, as indicated by the GI_24h_ value of 7.00%, which is comparable to the finding of 4.40% in a previous report of the administered TEV dose excreted in bile after 60 min [24]. After oral administration, the TEV was rapidly absorbed from the gastrointestinal tract and detected in the plasma from the first or second blood sampling time (5 or 15 min), reaching the *T*_max_ at 45 min. Once absorbed, the *t*_1/2_ of TEV was 115 min, and it was mainly eliminated through renal excretion, similar to in the intravenous study. The absolute *F* of TEV was 6.03%, and its GI_24h_ value was 30.8% of the administered TEV dose, suggesting a low absorption of TEV.

The mean arterial plasma concentration–time profiles and relevant pharmacokinetic parameters of TEV after oral administration of the SESS-TD crystal and TDF are also shown in Figure 2 and Table 2, respectively. After oral administration, of the SESS-TD crystal and TDF, the pharmacokinetic profiles of TEV were changed compared to those after oral administration of TEV; there were greater systemic exposures with a higher *C*_max_ (252 and 269% increases), greater AUCs (157 and 261% increases), smaller GI_24h_ values (67.8 and 65.0% decreases), and greater *F* values (157 and 272% increases) in the SESS-TD and TDF groups, respectively, than in the TDF group. However, the *t*_1/2_ values of TEV were comparable among the three groups; there was no significant change in TEV elimination with comparable CL values adjusted by the absolute *F* of each group (i.e., CL_oral_) in the TEV, the SESS-TD crystal, and TDF groups. Moreover, the estimated ‘*F*_unabs_’ values for the TEV, SESS-TD, and TDF groups were 30.4, 8.79, and 9.09%, respectively, from the following equations [23]:0.308 = ‘*F*_unabs_’ + (0.0700 × 0.0603) TEV
0.0991 = ‘*F*_unabs_’ + (0.0700 × 0.160) SESS-TD crystal
0.0106 = ‘*F*_unabs_’ + (0.0700 × 0.216) TDF
where 30.8% (7.00 and 6.03%), 9.91% (7.00 and 16.0%), and 10.6% (7.00 and 21.6%) were the oral GI_24h_, intravenous GI_24h_, and absolute *F*, respectively, of TEV (SESS-TD crystal and TDF) (Table 2).

### 3.3. TEV Pharmacokinetics after Administration of the SESS-TD Crystal and TDF Stored for 12 Months in Rats

The mean arterial plasma concentration–time profiles and relevant pharmacokinetic parameters of TEV after oral administration of the SESS-TD crystal and TDF, which were stored for 12 months, are shown in Figure 2 and Table 3, respectively. The mean arterial plasma concentration–time profiles and relevant pharmacokinetic parameters of TEV are also shown in Figure 2 and Table 3, respectively, as a positive control. In the stored SESS-TD crystal and TDF groups, the AUC values of TEV were greater (156 and 98.0% increase) and the GI_24h_ values of TEV were smaller (52.8 and 44.5% decrease) in the TEV group. However, the *t*_1/2_ and CL_oral_ values were comparable among the three groups. Moreover, the estimated ‘*F*_unabs_’ values for the TEV, SESS-TD, and TDF groups were 42.2, 19.3, and 23.0%, respectively, from the following equations [23]:0.425 = ‘*F*_unabs_’ + (0.0356 × 0.0895) TEV
0.201 = ‘*F*_unabs_’ + (0.0356 ×0.229) SESS-TD crystal
0.236 = ‘*F*_unabs_’ + (0.0356 × 0.177) TDF
where 42.5% (3.56 and 8.95%), 20.1% (3.56 and 22.9%), and 23.6% (3.56 and 17.7%) were the oral GI_24h_, intravenous GI_24h_, and absolute *F*, respectively, of TEV (SESS-TD crystal and TDF) (Table 3). These results indicate that the SESS-TD crystal enhanced TEV absorption even after being stored for 12 months. There was no difference between the stored SESS-TD crystal and TDF groups, and this pattern was similar to that seen with the SESS-TD crystal and TDF administration without storage.

## 4. Discussion

The pKa of TDF is 3.75, and it has a partition coefficient (log P) of 1.25. TDF is a salt form of TD that includes fumarate salt, which functions as a buffering agent to increase the pH of the drug, improving its solubility and absorption [25]. However, the SESS-TD crystal was designed to produce TD in a solid state, which also helps prevent the ease of the degradation of TDF in the presence of moisture [11]. As a follow-up to the study by An et al. (2017), the pharmacokinetic feasibility of the SESS-TD crystal was determined.

As shown in Appendix A, the percentages of remaining TD concentration in the rat plasma spiked with SESS-TD and TDF rapidly decreased by up to 11.9% and 3.73%, respectively, over time, and the TEV concentrations were increased. This result suggests that TD in SESS-TD and TDF might rapidly convert to TEV by esterase enzymes in rat plasma, as previously reported [26,27].

In the intravenous study, the *V*_ss_ of TEV (1917 mL/kg in Table 2) was higher than the reported values of plasma volume, which were 8.04 mL/189 g [28] and 7.8 mL/250 g [29], in the Sprague–Dawley rats. This suggests that the TEV was extensively distributed throughout the body. Due to the low plasma protein binding value of only 1.06%, the volume of distribution adjusted by the unbound fraction of TEV was similar to its *V*_ss_ calculated in Table 2. The main route of TEV elimination was through renal excretion, with the CL_R_ contributing to 65.7% of the CL (Table 2), indicating the active secretion of TEV via renal excretion. The estimated CL_R_ of TEV, considering its free fractions in plasma (CL_R_, fu), was 23.2 mL/min/kg. This was faster than the reported glomerular filtration rate (GFR), represented by the creatinine clearance, which was 5.24 mL/min/kg in rats [29], confirming the active secretion of TEV via renal excretion. The contribution of the gastrointestinal (including biliary) excretion of unchanged TEV to its CL_NR_ might be almost negligible, as the GI_24h_ was only 7.00% of the intravenous TEV dose (Table 2). Similarly, previous studies have reported that TEV is mainly eliminated through renal excretion, with low biliary excretion of TEV as the parent form into feces (i.e., 4.40% of biliary excretion for 60 min after oral administration) [24,30].

In oral studies, the *F*, AUC, and C_max_ are important tools in pharmacokinetics for characterizing the disposition of drugs in the body, assessing drug exposure, and predicting drug efficacy and safety. They are commonly used in drug development, dosing regimen optimization, and clinical practice to guide the use of medications in patients. In Table 2, the increased systemic exposures of TEV in the SESS-TD crystal and TDF groups were caused by the higher *C*_max_, greater AUC, smaller GI_24h_ values, and greater *F* values than in the TEV group. Specifically, the estimated ‘*F*_unabs_’ values for the TEV, SESS-TD, and TDF groups were 30.4, 8.79, and 9.09%, respectively (Table 3), indicating that TEV absorption was enhanced in the SESS-TD crystal and TDF groups. The absorption rates of TEV in the SESS-TD crystal and TDF groups were also faster, as indicated by their shorter *T*_max_ values than those in the TEV group. When orally administered with the 12-month stored TEV, SESS-TD, and TDF, the estimated ‘*F*_unabs_’ values for the TEV, SESS-TD, and TDF groups were 42.2, 19.3, and 23.0%, respectively (Table 3), indicating that TEV absorption was enhanced in the 12-month stored SESS-TD crystal and TDF groups. The absorption rates of TEV in the 12-month stored SESS-TD crystal and TDF groups were also faster, as indicated by their shorter *T*_max_ values than those in the 12-month stored TEV group.

In terms of elimination, the renal excretions of TEV were estimated based on the CL_oral_ as follows: (1) the CL_R_,_oral_ values of TEV were 27.6, 23.4, and 24.5 mL/min/kg in the TEV, SESS-TD crystal, and TDF groups, respectively; (2) the CL_R,fu,oral_ of TEV adjusted by the free fraction of TEV values from the TEV, SESS-TD, and TDF groups were 27.3, 23.2, and 24.2 mL/min/kg, respectively. All of the CL_R,fu,oral_ values of TEV were almost similar to the CL_R,fu_ values of TEV estimated in the intravenous study, indicating that active secretion was still the main renal excretion mechanism of TEV, and their values were comparable among the three groups (Table 2). Notably, both the SESS-TD crystal and TDF were primarily eliminated by renal excretion and had similar mechanisms of action [25,31]. Therefore, the greater AUC and absolute *F* values of TEV in the SESS-TD crystal and TDF groups than in the TEV group could have been due to the increased absorption of TEV rather than changes in the TEV elimination among the three groups.

In both the stored SESS-TD crystal and TDF groups, a greater AUC of TEV was observed than in the TEV group. In addition, no significant difference was observed between the stored SESS-TD crystal and TDF groups. These findings suggest that the stored SESS-TD crystal did not deteriorate even after being stored at room temperature for 12 months.

In conclusion, the pharmacokinetic profile of TEV after SESS-TD crystal administration was found to be comparable to the marketed form of TDF. Notably, the absolute *F* of TEV after SESS-TD crystal administration was improved, indicating increased absorption that was similar to that of TDF. Our findings suggest that the pharmacokinetic feasibility of SESS-TD crystal is sufficient to replace TDF, and that SESS-TD crystal has the potential to be used as an API for TEV instead of TDF.

## Figures and Tables

**Figure 1 pharmaceutics-15-01392-f001:**
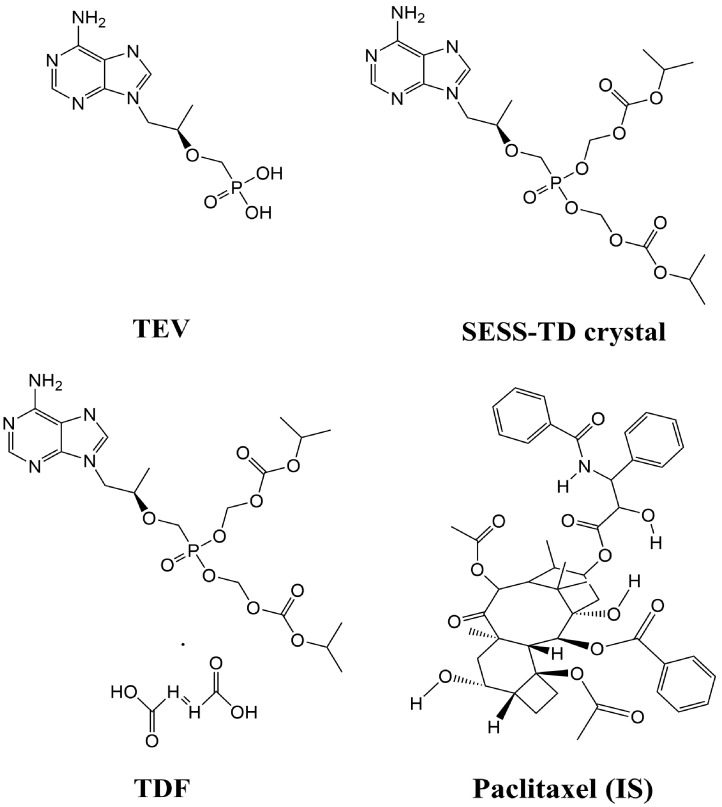
Structure of TEV, SESS-TD crystal, TDF, and paclitaxel (IS).

**Figure 2 pharmaceutics-15-01392-f002:**
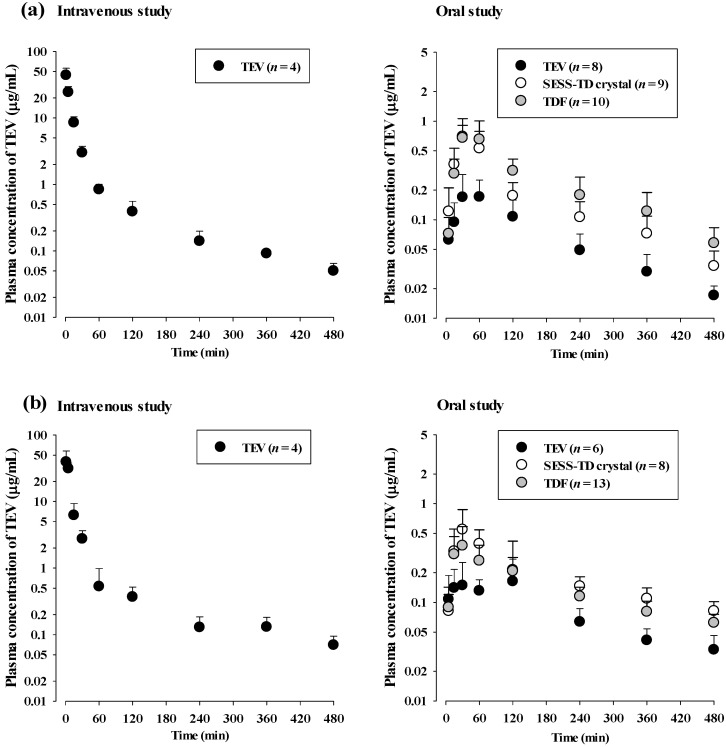
(**a**) Plasma concentration of TEV after intravenous administration of TEV (20 mg/kg) and oral administration of TEV, SESS-TD crystal, and TDF (as 20 mg/kg of TEV) to rats. (**b**) Plasma concentration of TEV after oral administration of the SESS-TD crystal and TDF stored for 12 months (as 20 mg/kg of TEV) to rats. As positive control, plasma concentration of TEV after intravenous and oral administration of TEV (20 mg/kg) without storage.

**Table 1 pharmaceutics-15-01392-t001:** Physicochemical properties of TEV, SESS-TD crystal, and TDF.

	TEV	SESS-TD Crystal	TDF
Chemical formula	C_9_H_14_N_5_O_4_P [13]	C_19_H_30_N_5_O_10_P [11,14]	C_23_H_34_N_5_O_14_P [4]
Formula weight (amu)	287.21 [13]	519.45 [11]	635.52 [11]
Crystal system	-	Orthorhombic [11]	TD/FA 1:1 salt [11]
pKa	3.75 [15]	-	3.75 [11]
log P	1.3 [15]	0.68 [11]	1.25 [11]
Solubility (mg/mL)			
distilled water	5.20 [15]	5.88 [14]	6.20 [14]
pH 1.2	43.4 [16]	34.9 [14]	35.0 [14]
pH 4.0	-	5.12 [14]	4.88 [14]
pH 6.8	-	6.91 [14]	6.45 [14]
Stability (%)			
15% relative humidity	-	100 [11]	99.90 [11]
35% relative humidity	-	100 [11]	99.67 [11]
55% relative humidity	-	99.99 [11]	99.44 [11]
75% relative humidity	-	99.98 [11]	99.21 [11]
95% relative humidity	-	99.96 [11]	98.60 [11]

**Table 2 pharmaceutics-15-01392-t002:** Pharmacokinetic parameters of TEV after intravenous administration of TEV (20 mg/kg) and oral administration of TEV, the SESS-TD crystal, and TDF (as 20 mg/kg of TEV) to rats.

Parameters	Intravenous			Parameters				Oral					
	TEV (*n* = 4)				TEV (*n* = 8)			SESS-TD Crystal (*n* = 9)			TDF (*n* = 10)		
Body weight (g)	301	±	8.54	Body weight (g)	311	±	9.91	317	±	20.6	319	±	22.6
AUC_0–t_ (μg min/mL)	554	±	35.4	AUC_0–t_ (μg min/mL) *	30.8	±	11.3	79.0	±	17.0	111	±	34.2
AUC_0–inf_ (μg min/mL)	564	±	37.6	AUC_0–inf_ (μg min/mL) *	34.0	±	11.6	87.5	±	19.4	127	±	41.7
AUMC_0–t _ (μg min^2^/mL)	20,296	±	4017	AUMC_0–t_ (μg min^2^/mL) *	4411	±	2122	10,556	±	3265	17,554	±	6575
AUMC_0-inf _ (μg min^2^/mL)	27,577	±	3980	AUMC_0–inf _ (μg min^2^/mL) *	6264	±	2543	16,496	±	6378	29,848	±	16,997
*t*_1/2_ (min)	166	±	25.9	*t*_1/2_ (min)	115	±	20.9	153	±	46.3	167	±	76.2
*V*_ss_ (mL/kg)	1917	±	47.2	*C*_max_ (μg/mL) *	0.234	±	0.100	0.823	±	0.299	0.865	±	0.210
CL (mL/min/kg)	35.6	±	2.37	*T*_max_ (min) ^a^	45 (15–60)			30 (15–60)			45 (15–60)		
CL_R_ (mL/min/kg)	23.4	±	1.51	CL/*F* (mL/min/kg) *	667	±	275	239	±	55.3	177	±	66.6
CL_NR_ (mL/min/kg)	12.2	±	1.00	CL_oral_ (mL/min/kg)	40.3	±	16.6	37.1	±	8.58	39.6	±	14.9
MRT_0-t_ (min)	36.5	±	5.85	MRT_0-t_ (min)	138	±	27.2	134	±	31.2	156	±	20.8
MRT_0-inf_ (min)	48.7	±	3.82	MRT_0-inf_ (min)	183	±	37.2	188	±	32.1	225	±	68.6
Ae_0–24h_ (%)	65.9	±	1.10	Ae_0–24h_ (%)	66.9	±	8.15	62.3	±	54.9	61.3	±	11.4
GI_24h_ (%)	7.00	±	3.79	GI_24h_ (%) *	30.8	±	8.48	9.91	±	4.40	10.6	±	5.05
				Absolute *F* (%)	6.03			16.0			21.6		
				Relative *F* (%)				266			359		
				*F* _unabs_	30.4			8.79			9.09		

^a^ Data are reported as median (ranges). * The TEV group was significantly different (*p* < 0.05) from the SESS-TD crystal and TDF groups.

**Table 3 pharmaceutics-15-01392-t003:** Pharmacokinetic parameters of TEV after oral administration of the SESS-TD crystal and TDF (as 20 mg/kg of TEV) stored for 12 months. Pharmacokinetic parameters of TEV after intravenous and oral administration of 20 mg/kg of TEV to rats.

Parameters	Intravenous			Parameters				Oral					
	TEV (*n* = 4)				TEV (*n* = 6)			SESS-TD Crystal (*n* = 8)			TDF (*n* = 13)		
Body weight (g)	323	±	28.7	Body weight (g)	304	±	25.3	312	±	33.5	318	±	28.6
AUC_0–t_ (μg min/mL)	516	±	86.8	AUC_0–t_ (μg min/mL) *	38.8	±	13.5	87.2	±	27.0	68.2	±	21.0
AUC_0–inf_ (μg min/mL)	533	±	84.1	AUC_0–inf_ (μg min/mL) *	47.8	±	15.9	122	±	35.9	94.6	±	23.1
AUMC_0–t _ (μg min^2^/mL)	21,127	±	6633	AUMC_0–t _ (μg min^2^/mL) *	6544	±	2332	15,305	±	3943	11,901	±	2561
AUMC_0–inf _ (μg min^2^/mL)	33,997	±	10,787	AUMC_0–inf _ (μg min^2^/mL) *	13,002	±	4764	48,469	±	22,049	36,955	±	17,401
*t*_1/2_ (min)	170	±	28.8	*t*_1/2_ (min)	174	±	40.4	291	±	97.8	281	±	100
*V*_ss_ (mL/kg)	2549	±	1275	*C*_max_ (μg/mL) *	0.222	±	0.116	0.554	±	0.286	0.394	±	0.204
CL (mL/min/kg)	38.2	±	6.18	*T*_max_ (min) ^a,+^	90 (15–120)			30 (30–90)			30 (15–120)		
CL_R_ (mL/min/kg)	27.3	±	5.24	CL/*F* (mL/min/kg) *	464	±	167	179	±	60.9	222	±	48.3
CL_NR_ mL/min/kg)	10.9	±	2.16	CL_oral_ (mL/min/kg)	41.5	±	15.0	40.9	±	13.9	39.4	±	8.57
MRT_0–t_ (min)	41.4	±	12.1	MRT_0–t_ (min)	169	±	18.9	177	±	11.9	279	±	26.1
MRT_0–inf_ (min)	65.1	±	24.5	MRT_0–inf_ (min)	272	±	36.1	390	±	110	389	±	137
Ae_0–24 h_ (%)	71.4	±	4.29	Ae_0–24 h_ (%)	67.7	±	8.40	69.6	±	10.3	64.7	±	9.91
GI_24 h_ (%)	3.56	±	2.72	GI_24 h_ (%) *	42.5	±	10.1	20.1	±	9.20	23.6	±	13.7
				Absolute *F* (%)	8.95			22.9			17.7		
				Relative *F* (%)				256			198		
				*F* _unabs_	42.2			19.3			23.0		

^a^ Data are reported as median (ranges). * The TEV group was significantly different (*p* < 0.05) from the SESS-TD crystal and TDF groups. ^+^ The TEV group was significantly different (*p* < 0.05) from the SESS-TD crystal group, but not from TDF group.

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
