# Peer review of "Pharmacokinetic Feasibility of Stability-Enhanced Solid-State (SESS) Tenofovir Disoproxil Free Base Crystal"

_pharmaceutics, 2023, doi:10.3390/pharmaceutics15051392_

Round 1

Reviewer 1 Report

In this manuscript, the authors presented pharmacokinetic studies of tenofovir, tenofovir disoproxil and a special formulation (SESS). The study is simple and straightforward, but very little new or innovative information was provided. If the new formulation is novel, the authors should focus on formulation development.

Some other comments:

1. Various sample sizes have been used for different treatment groups.

2. Body weight is very different in oral dose (12 months group).

3. The discussion about Funabs (lines 265 to 268) should be in Method and Results.

Author Response

Answers for the reviewer 1’s comments:

In this manuscript, the authors presented pharmacokinetic studies of tenofovir, tenofovir disoproxil and a special formulation (SESS). The study is simple and straightforward, but very little new or innovative information was provided. If the new formulation is novel, the authors should focus on formulation development.

→ Since the formulation development process and in vitro physicochemical property of stability-enhanced solid state (SESS) tenofovir disoproxil free base crystal (SESS-TD crystal) have been reported by An et al (2017) and KR1020140151114A (November 03.KR101548724). Thus, the evaluation of in vivo pharmacokinetic feasibility of SESS-TD crystal was focused in this manuscript.

Some other comments:

  1. Various sample sizes have been used for different treatment groups.

→ As you commented, various sample sizes for each group have been used due to the limitations in TEV, SESS-TD crystal and TDF amounts. The same compounds of TEV, SESS-TD crystal and TDF which were used in formulation process and in vitro test of physicochemical properties were administered to rats in this study. Since TEV should be divided and used for intravenous and oral administration studies, the numbers of rats for TEV group (i.e., intravenous and oral groups of TEV, respectively) were unfortunately small compared to those in SESS-TD crystal and TDF groups. In other words, the total sum number of rats for intravenous and oral administration of TEV were twelve and ten rats shown in Figure 2 and Tables 2 and 3.

  1. Body weight is very different in oral dose (12 months group).

→ Body weights in Table 3 were the initial body weights of rats when they arrived in the animal center, which were written by mistake. Body weights in the experiment day were checked again and corrected in the revised manuscript.

  1. The discussion about Funabs (lines 265 to 268) should be in Method and Results.

→ The method for the estimation of Funabs was added in the method section. Also the Funabs values were added in the Tables 2 and 3, which were described in the result section.

Reviewer 2 Report

The only issue I have with this paper is there is no Table of the physicochemical properties of TEV, SESS-TD and TDF. Make a table of pkA, solubility, dissolution

logP etc show the differences there needs to be a better link between this and the PK results. Perhaps even show a graph of dissolution or solubility changes. The main finding is F, AUC and CMax and the underlying reason for this needs to be explored much better

Author Response

Answers for the reviewer 2’s comments:

The only issue I have with this paper is there is no table of the physicochemical properties of TEV, SESS-TD and TDF.

→ As you commented, Table 1 was added and the physicochemical properties of TEV, SESS-TD and TDF were listed in Table 1.

Make a table of pkA, solubility, dissolution logP etc show the differences there needs to be a better link between this and the PK results. Perhaps even show a graph of dissolution or solubility changes.

→ As you commented, the physicochemical properties of TEV, SESS-TD and TDF were listed in Table 1. Unfortunately, several information could not be added in Table 1: for example, the compounds used in this study were powder and their dissolution were not measured.

The main finding is F, AUC and CMax and the underlying reason for this needs to be explored much better

→ The underlying reasons for the changes of pharmacokinetic parameters such as F, AUC and Cmax of TEV in SESS-TD crystal compared to those of TEV were additionally described in the result and discussion section, as you suggested.

Answers for the reviewer 2’s comments:

The only issue I have with this paper is there is no table of the physicochemical properties of TEV, SESS-TD and TDF.

→ As you commented, Table 1 was added and the physicochemical properties of TEV, SESS-TD and TDF were listed in Table 1.

Make a table of pkA, solubility, dissolution logP etc show the differences there needs to be a better link between this and the PK results. Perhaps even show a graph of dissolution or solubility changes.

→ As you commented, the physicochemical properties of TEV, SESS-TD and TDF were listed in Table 1. Unfortunately, several information could not be added in Table 1: for example, the compounds used in this study were powder and their dissolution were not measured.

The main finding is F, AUC and CMax and the underlying reason for this needs to be explored much better

→ The underlying reasons for the changes of pharmacokinetic parameters such as F, AUC and Cmax of TEV in SESS-TD crystal compared to those of TEV were additionally described in the result and discussion section, as you suggested.

Reviewer 3 Report

Comment

Date: 07-04-2023

Manuscript ID: pharmaceutics-2343339

You et al. addressed very interesting findings in their research article entitle as “Pharmacokinetic feasibility of stability-enhanced solid state 2(SESS) tenofovir disoproxil free base crystal”. The work is interesting, well designed, and informative to reader working in the domains. However, I recommend few suggestions before publication.

Comment 1: In abstract, the sentence “An et al. (2017) designed a stability-enhanced 16solid state-TD free base crystal (SESS-TD crystal), which showed improved solubility and stability 17compared with TEV” must be removed. Abstract section needs to be revised by adding finding in quantitative way and comparing with control.       

Comment 2: Introduction section is inconsistent and short. I appreciate with the content. However, various information are missing such as dose, drug instability at different pH and aqueous systems, physicochemical properties of the drug (pka, logP, hydrogen bonding acceptor and donor counts, and aqueous solubility reported), major metabolites in human bodies and strategies adopted to circumvent these issues as reported in literatures.     

Comment 3: In material section, authors should write [percent purity of the drug and known impurities. Material section must be expanded for missing other chemicals used in the study. Moreover, the source of material is missing like city, state, and country. 

Comment 4: In section 2.3, what was the total run time, and sampling volume? Moreover, authors should include limit of detection, and lower limit of quantification. What was the concentration of stock solution of calibration curve and regression equation in the studied solvent? How was the blood sampling strategy from rat? Please provide in an illustrative way. In the sentence “For the intravenous study, a dose of 20 mg (4 mL)/kg of TEV (dissolved in distilled water) was intravenously administered to rats.”. Practical solubility of the drug in water is less than 1 mg/ml. How did authors deliver parenterally? This will cause embolism or the blockage of fine blood capillaries. Please explain.        

Comment 5: In figure 2, I found that figure 2a-b (IV) has 10 time points whereas figure 2a-b (oral) depicted only nine time points. Why did authors vary time points if compared? I suggest to keep the same time points in both cases while comparing. Or provide reason for the selected time points.  

Comment 6: In table 1, few pharmacokinetics parameters are missing. These are Tmax, AUMC and MRT. Moreover, AUC for the fixed or infinity? These must be added and revised as per suggestion.

Comment 7: In introduction section, there should be explanation for the concept of prodrug synthesis and mechanistic way in vivo degradation. Moreover, authors must present an illustrative or diagrammatic presentation of prodrug synthesis and its degradation at different pH after oral administration and IV.  

Author Response

Answers for the reviewer 3’s comments

You et al. addressed very interesting findings in their research article entitle as “Pharmacokinetic feasibility of stability-enhanced solid state 2(SESS) tenofovir disoproxil free base crystal”. The work is interesting, well designed, and informative to reader working in the domains. However, I recommend few suggestions before publication.

Comment 1: In abstract, the sentence “An et al. (2017) designed a stability-enhanced solid state-TD free base crystal (SESS-TD crystal), which showed improved solubility and stability compared with TEV” must be removed. Abstract section needs to be revised by adding finding in quantitative way and comparing with control.

→ As you mentioned, the sentence “An et al. (2007) designed a stability-enhanced solid state TD free base crystal (SESS-TD crystal), ….” was removed. The quantitative change of SESS-TD crystal compared to TEV was added in the abstract section.

Comment 2: Introduction section is inconsistent and short. I appreciate with the content. However, various information are missing such as dose, drug instability at different pH and aqueous systems, physicochemical properties of the drug (pka, logP, hydrogen bonding acceptor and donor counts, and aqueous solubility reported), major metabolites in human bodies and strategies adopted to circumvent these issues as reported in literatures.

→ As you commented, the introduction section was revised. However, this study has focused on the pharmacokinetic feasibility of SESS-TD because the change of physiochemical properties of SESS-TD crystal has been described in the previous study by An et al. (2017).

Comment 3: In material section, authors should write percent purity of the drug and known impurities. Material section must be expanded for missing other chemicals used in the study. Moreover, the source of material is missing like city, state, and country.

→ As you pointed out, the purity percentages and missing other chemicals used in this study were added in the material section. The sources of materials were also added in the material section.

Comment 4: In section 2.3, what was the total run time, and sampling volume? Moreover, authors should include limit of detection, and lower limit of quantification. What was the concentration of stock solution of calibration curve and regression equation in the studied solvent? How was the blood sampling strategy from rat? Please provide in an illustrative way. In the sentence “For the intravenous study, a dose of 20 mg (4 mL)/kg of TEV (dissolved in distilled water) was intravenously administered to rats.”. Practical solubility of the drug in water is less than 1 mg/ml. How did authors deliver parenterally? This will cause embolism or the blockage of fine blood capillaries. Please explain.

→ The total run time was 7 min described in the result section. In sample preparation process, 50 µL of biological sample was deproteinized by 200 µL of acetonitrile, vortexed, and centrifuged for 10 min and 13000 rpm. A 100 µL of the supernatant was transferred to vial, and then a 10 µL was injected into the column. The limit of detection and lower limit of quantification were clearly added. These descriptions in section 2.3 were revised considering your comment.   

In calibration curves, 20 mg/mL of TEV stock solution in DMSO was serially diluted to 2, 1, 0.5, 0.2, 0.1, 0.05, 0.01, 0.0025, 0.001 mg/mL by methanol. A 20 mg/mL of stock solution and various concentrations of working solutions of TEV were added to drug-free plasma samples to achieve 0.01, 0.025, 0.1, 0.5, 1, 2, 5, 10, 20, or 200 µg/mL of plasma standard samples, respectively. The urine and GI standard samples were also made by the same method as the plasma standard samples. The calibration curve ranges for TEV in biological standard samples of plasma, urine, and GI were 0.01–200 µg/mL (y = 472 x + 1645, R2= 0.999), 1–20 µg/mL (y = 17.3 x + 1143, R2= 1.00), and 0.1–20 µg/mL (y = 59.5 x + 1065, R= 0.998), respectively. These descriptions in section 2.3 were revised considering your comment.

In blood sampling, total blood sampling volume was 0.96–1.08 mL (i.e., 0.12 mL of blood was collected for 9 and 8 times in intravenous and oral study, respectively), which were acceptable method considering that 1.7-2.1 ml/300 g without fluid supplement and 2.5-3.2 ml/300 g with fluid supplement are acceptable in rats. The description of blood sampling method was revised illustratively in section 2.5.

In administered solution of TEV, 20 mg/4 mL of TEV was dissolved in the distilled water considering the solubility of 5.20 mg/mL in the distilled water (An et al., 2017; Spinks et al., 2017).

Comment 5: In figure 2, I found that figure 2a-b (IV) has 10 time points whereas figure 2a-b (oral) depicted only nine time points. Why did authors vary time points if compared? I suggest to keep the same time points in both cases while comparing. Or provide reason for the selected time points.

→ In the intravenous study, blood sampling at 1 min was conducted to measure TEV concentration at early sampling as much as possible after intravenous administration. Also considering the absorption rate (i.e., Cmax and Tmax) TEV from the literature, the blood sampling points were determined. However, the plasma concentration-time curve of TEV in figure 2 and pharmacokinetic parameters of TEV in Tables 2 and 3 were revised using the same blood sampling data as you suggested in the revised manuscript.

Comment 6: In table 1, few pharmacokinetics parameters are missing. These are Tmax, AUMC and MRT. Moreover, AUC for the fixed or infinity? These must be added and revised as per suggestion.

→ Table 1 was moved to Table 2 in the revised manuscript. The pharmacokinetic parameters in Tables 2 and 3 were revised as you suggested.

Round 2

Reviewer 1 Report

No additional comments.

Reviewer 3 Report

Authors considered the suggested comments and they uploaded revised version suitable for publication. I recommend for publication of the revised manuscript.